# SARS-CoV-2 N-Antigen Quantification in Respiratory Tract, Plasma and Urine: Kinetics and Association with RT-qPCR Results

**DOI:** 10.3390/v15051041

**Published:** 2023-04-24

**Authors:** Delphine Parraud, Anne-Lise Maucotel, Maude Bouscambert, Florence Morfin, Laurent Bitker, Christian Chidiac, Nathalie De Castro, Emilie Frobert, Alexandre Gaymard

**Affiliations:** 1Laboratoire de Virologie, Institut des Agents Infectieux, Laboratoire Associé Au Centre National de Référence des Virus des Infections Respiratoires, Hospices Civils de Lyon, F-69317 Lyon, France; delphine.parraud@chu-lyon.fr (D.P.);; 2CIRI, Centre International de Recherche en Infectiologie, Team VirPath, Univ Lyon, Inserm, U1111, Université Claude Bernard Lyon 1, CNRS, UMR5308, ENS de Lyon, F-69007 Lyon, France; 3Service de Médecine Intensive-Réanimation, Hôpital de la Croix Rousse, Hospices Civils de Lyon, 103 Grande Rue de la Croix Rousse, CEDEX 04, F-69317 Lyon, France; 4Service des Maladies Infectieuses et Tropicales, Hospices Civils de Lyon, Groupement Hospitalier Nord, Université Claude Bernard Lyon 1, F-69317 Lyon, France; 5Service des Maladies Infectieuses et Tropicales, Hôpital Saint-Louis, APHP, F-75010 Paris, France

**Keywords:** SARS-CoV-2, antigen detection, quantitative assay, urine

## Abstract

Qualitative SARS-CoV-2 antigen assays based on immunochromatography are useful for mass diagnosis of COVID-19, even though their sensitivity is poor in comparison with RT-PCR assays. In addition, quantitative assays could improve antigenic test performance and allow testing with different specimens. Using quantitative assays, we tested 26 patients for viral RNA and N-antigen in respiratory samples, plasma and urine. This allowed us to compare the kinetics between the three compartments and to compare RNA and antigen concentrations in each. Our results showed the presence of N-antigen in respiratory (15/15, 100%), plasma (26/59, 44%) and urine (14/54, 28.9%) samples, whereas RNA was only detected in respiratory (15/15, 100%) and plasma (12/60, 20%) samples. We detected N-antigen in urine and plasma samples until the day 9 and day 13 post-inclusion, respectively. The antigen concentration was found to correlate with RNA levels in respiratory (*p* < 0.001) and plasma samples (*p* < 0.001). Finally, urinary antigen levels correlated with plasma levels (*p* < 0.001). Urine N-antigen detection could be part of the strategy for the late diagnosis and prognostic evaluation of COVID-19, given the ease and painlessness of sampling and the duration of antigen excretion in this biological compartment.

## 1. Introduction

The COVID-19 pandemic, caused by the SARS-CoV-2 virus, has been responsible for hundreds of millions of cases and millions of deaths worldwide since it began in Wuhan, China, in 2020 [1].

The current gold standard for COVID diagnosis is viral RNA detection using real-time reverse transcription-PCR (RT-PCR) [2]. Diagnosis is commonly performed on respiratory specimens because RT-PCR has been shown to be less sensitive in other types of specimens (e.g., plasma, urine and feces) [3].

In addition to molecular testing, there is still a need to develop alternative tools for SARS-CoV-2 detection, given the cost of reagents, the laboratory capacity required to perform molecular biology assays, and the length of analysis. One such tool is viral antigen testing, which mainly detects the nucleocapsid (NP) protein of SARS-CoV-2. Since 2020, many of these qualitative immunochromatographic assays have been commercialized, as antigen testing is quick and easy to perform, and does not require specialized facilities and equipment. However, many of these assays have shown lower sensitivity than RNA detection in respiratory specimens: in fact, a meta-analysis showed that the sensitivity of antigen assays depends on viral load, time after symptom onset and also the brand of the assay [4,5]. Despite their lower sensitivity, antigen assays have a high specificity in symptomatic and asymptomatic patients [4], which could allow clinicians to quickly classify their patients and help with triage.

Another way of exploiting antigen testing could be to use quantitative assays: indeed, antigen concentration in a respiratory sample correlates to viral load [6] and can help to define the stage of infection [7]. In plasma samples, SARS-CoV-2 antigen concentration also appeared to correlate with the viral load in respiratory samples [8], and quantification of plasmatic antigen can provide important information for patients’ prognostication [9,10]. Recently, antigen detection in urine was shown to be frequent during the first two weeks after symptom onset in patients with a positive respiratory RT-PCR [8], and its quantification using a COVID-QUANTO^®^ kit (AAZ) showed higher positivity rates in ICU patients. However, the kinetics of clearance in different compartments and the quantification of SARS-CoV-2 antigen in these compartments need to be investigated simultaneously.

Using the COVID-QUANTO^®^ CE-IVD ELISA microplate assay (AAZ), we measured and quantified plasmatic, urinary and respiratory levels of N-antigen. Our aim was to compare the kinetics between viral load and antigen levels in each compartment to better understand the physiopathology of NP protein excretion.

## 2. Materials and Methods

### 2.1. Patients and Samples

One hundred and thirty-one samples from 26 patients hospitalized in Lyon between March and April 2020 were retrospectively tested for SARS-CoV-2 RNA and N-antigen quantification. Patients were selected if multiple respiratory, urine and plasma samples were available during follow-up. All patients were included in the French COVID cohort (clinicaltrials.gov NCT04262921) with ethics approval from the CPP IDF VI (ID RCB: 2020-A00256-33). The median delay between the first positive SARS-CoV-2 RT-PCR and enrolment was 3 days (min 0 and max 11 days). Samples were collected between day 1 and day 21 after enrolment and were frozen at −80 °C for conservation. For our data analysis, days after inclusion was used as the time scale. All respiratory and plasma samples were tested for viral RNA and antigen quantification, except one sample of each type that was tested for RNA quantification only due to insufficient volume. For urine samples, due to the low RNA detection rate described in previous studies, all samples were tested for antigens but only positive samples were tested for viral RNA.

### 2.2. RT-PCR Assay

Viral RNA was detected in respiratory, plasma and urine samples using IP2/IP4 primers/probes as previously described [11]. Viral load determination was performed on all samples by RNA extraction on the EMAG^®^ platform (bioMerieux, Marcy-l’Étoile, France). The SARS-CoV-2 load was determined by quantitative RT-PCR according to a calibrated in-house plasmid scale using the IP2-IP4 target developed by the Institut Pasteur (Paris, France). The amplification protocol was performed using QuantStudio 5 rtPCR Systems (Thermo Fisher Scientific, Waltham, MA, USA) [11]. The absence of inhibitors in the specimen was checked by using the RNA Internal Control R-GENE^®^ kit (Argene_BioMérieux, Marcy-l’Étoile, France) on each sample. We expressed SARS-CoV-2 viral load in RNA copies/mL.

### 2.3. N-Antigen Detection

Antigen detection was performed manually using the COVID-QUANTO^®^ CE-IVD microplate ELISA assay (AAZ LMB, Boulogne-Billancourt, France) according to the manufacturer’s instructions. Briefly, 50 µL of sample and 50 µL of biotinylated anti-SARS-CoV-2 N-antibodies were added to 96-well microplate wells coated with anti-SARS-CoV-2 N-antibodies. The plates were then incubated at 37 °C for 60 min, and then washed and incubated a second time at 37 °C for 30 min. After a further wash, 50µL of horseradish peroxidase-conjugated streptavidin and 50ìL of a solution containing the substrate (3,3′,5,5′-tetramethylbenzidine (TMB)) were added prior to 15 min incubation at 37 °C. A stop solution was then added and the OD was measured at 450 nm. A standard solution of recombinant N-antigen provided in the assay was used to determine the concentration in each sample. The threshold for a positive result was 3 pg/mL and all samples below this limit were considered negative, as recommended by the manufacturer.

### 2.4. Statistics and Software

Data were analyzed using RStudio Team (2020, PBC, Boston). Correlations were calculated by using Spearman’s method.

## 3. Results

A total of 60 plasma, 54 urine and 17 respiratory samples were collected. The clinical and biological characteristics of the patients are summarized in Appendix A. The patients were predominantly men (*n* = 25/26, 96%), with a median age of 67 ([27; 86]) Comorbidities frequently encountered were obesity (8/26, 31%), hypertension (8/23, 35%) and chronic cardiac diseases (7/25, 28%). Only one patient presented with chronic kidney disease (1/25, 4%). A total of 25/26 (96%) patients were admitted to the intensive care unit (ICU), including 24 males and 1 female. Twelve of the ICU patients died during hospitalization. The biological results did not show any particular impairment of kidney function, with mild levels of creatinine (median = 81 µmol/L [70; 209]) and urea (median = 7.0 mmol/L, [3.0; 15.1]).

The RNA and N-antigen detection and quantification results are summarized in Figure 1 and Table 1.

For respiratory samples, 15/16 (94%) were positive for RNA on day 1, of which, 14 were also positive for N-antigen (Figure 1), with high rates for both biological markers (Table 1). One sample tested positive for N-antigen alone (184 pg/mL) with a negative RT-PCR result. One sample was tested on day 9 and was positive for both RNA and antigen, respectively, reaching 335 copies/mL and 3.1 pg/mL.

For plasma samples, RNA detection showed decreasing rates between day 1 (2/3, 66% samples detected positive), day 3 (8/15, 53%), day 7 (2/18, 11%), and then days 13 and 21 with no positive samples. Interestingly, plasma N-antigen was detected for longer than viral RNA, with 7/15 (46.6%) samples positive until day 13 (Figure 1). Of the 54 urine samples tested for antigen detection between day 1 and day 21, 14 were found positive: 3/3 (100%) on day 1, 8/12 (67%) on day 3 and 3/18 (17%) on day 9, with concentrations ranging from 3.94 to 1657 pg/mL (Table 1). These 14 samples were tested for RNA detection, and none were positive.

On day 1, antigen quantification was higher in respiratory samples (median 2005 pg/mL [min = 0 max = 17,500]) compared to plasma samples (919 pg/mL [12; 1412]) and urine samples (92 pg/mL [38; 1651]) (Figure 2A–C). These results are consistent with a viral RNA detection rate of up to 10^6^ copies/mL in the respiratory tract and only up to 917 copies/mL in plasma and undetectable in urine (Figure 2D,E). There was no clear trend in antigen levels or viral load according to deceased/not deceased status in any compartment (Figure 2). On day 1, all deceased patients had antigen levels above 1000 pg/mL in the respiratory tract (Figure 2A), whereas only half (3/6) of the surviving patients reached this level.

Antigen levels function of RNA quantification in plasma and respiratory samples are represented in Figure 3A,B. Plasma and respiratory antigen quantification rates were positively correlated with RNA quantification (ρ = 0.6167943; *p* < 0.001 and ρ = 0.8024581; *p* < 0.001, respectively). The urinary antigen function of plasma antigen is represented in Figure 3C. We found a strong correlation between these parameters (ρ = 0.821357; *p* < 0.001).

## 4. Discussion

Our work consisted of the detection and quantification of SARS-CoV-2 N-antigen in three biological compartments: the respiratory tract, plasma and urine.

Plasma N-antigen has previously been shown to be associated with disease severity and important patient outcomes, such as the increased risk of pulmonary exacerbation and delayed hospital discharge [9,12]. This quantification may be useful to better identify which patients have a better prognosis and are less likely to develop a severe form of the disease [13].

In our study, the N-antigen quantification correlated positively with the amount of RNA detected in both the respiratory and plasma samples. Interestingly, the correlation was stronger in the respiratory tract than in plasma. In total, 15/17 (88%) respiratory samples were positive for both assays, and none were double-negatives. In comparison 12/60 (20%) plasma samples were double-positive and 20/60 (33%) were double-negative. It is of note that 21/60 (35%) of the plasma samples were positive for N-antigen only, suggesting that the kinetics of antigen excretion and clearance may be different from those of the viral particle. Indeed, plasmatic N-antigen seemed to follow the same kinetics as RNA levels but was detected for longer than viral RNA (maximum 13 days for antigen versus 9 days for RNA). We attempted to link plasma N-antigen concentration with the risk of death: on day 1 of inclusion, all deceased patients had antigen levels >1000 pg/mL, but on day 3, 4/6 patients above this rate are patients who survived. This could mean that this rate of 1000 pg/mL can be used in the early stages of the infection but may be less useful as the infection progresses. This rate was already used as an early prognostic discriminator in the observational study of Wick et al. [13], with an NPV of 93% for a worse WHO original scale over time.

Regarding the urinary compartment, the urinary excretion of N-antigen was correlated with its plasmatic concentration. In our work, as already described, the detection of SARS-CoV-2 RNA in urine was negative [14], meaning that there is no viral but only antigenic excretion in the urinary compartment. Urinary N-antigen detection lasted until day 9 post-inclusion in 17% of patients tested (3/18), despite the negative detection of SARS-CoV-2 RNA in all urine samples. Regarding the quantitative results, urinary samples can contain a high level of N-antigen (Figure 2C). Our results are in agreement with Veyrenche et al. [8], with the detection of N-antigen at a higher concentration during the first week of disease onset and decreasing during the following weeks. As it is an easy and painless sample to obtain, antigen detection in urine could be of interest as an alternative to usual diagnostic techniques, and to help triage patients in the emergency department for example.

These data also suggest the potential contribution of urinary N-antigen detection and quantification to monitor patients and predict severe outcomes. Indeed, as plasmatic antigen detection is a marker of clinical deterioration, urinary antigen detection could be considered as a reflection of plasmatic excretion. A similar study has been conducted in patients infected by dengue virus (DENV), showing antigenic detection in urine may be related to the severity of infection [15,16]. In another microbiological field, antigenic urinary detection is already largely used to help diagnose bacterial community-acquired pneumonia induced by Legionella pneumophila and Streptococcus pneumoniae, allowing to adapt the patient’s care [17].

A limitation of our study is that we did not compare the excretion of SARS-CoV-2 antigen over time in outpatients or at least in less severe cases, because in this study, almost all of our patients were admitted to the ICU during hospitalization, and 12 deaths occurred. These patients were selected according to the availability of urine and plasma samples, which happened to be collected mainly from ICU patients.

It would also be interesting to assess the sensitivity and specificity of COVID-QUANTO^®^ CE-IVD ELISA microplate assay (AAZ LMB, Boulogne-Billancourt, France) in urine, by including a control group.

In summary, this study has demonstrated the presence of SARS-CoV-2 N-antigen in several biological compartments. In the respiratory tract and plasma, N-antigen correlates with viral RNA detection. In terms of kinetics, N-antigen can be detected in plasma longer than RNA. Even when no viral RNA is detected in urine, N-antigen can reach high levels in this compartment, and its concentration correlates with plasmatic N-antigen. Prolonged antigenic shedding in both plasma and urine could be used in a diagnostic strategy for severe forms of COVID-19. Indeed, if upper respiratory samples are negative for RNA detection, lower respiratory samples may be considered as the matrix of choice for diagnosis. However, urine sampling is much less painful and complicated than lower respiratory tract sampling and could be an interesting adjunct to COVID-19 diagnosis.

With these data, we also presented the capacity of the COVID-QUANTO^®^ CE-IVD ELISA microplate assay to detect N-antigen in different types of samples and its potential contribution to non-invasive COVID-19 diagnosis.

## Figures and Tables

**Figure 1 viruses-15-01041-f001:**
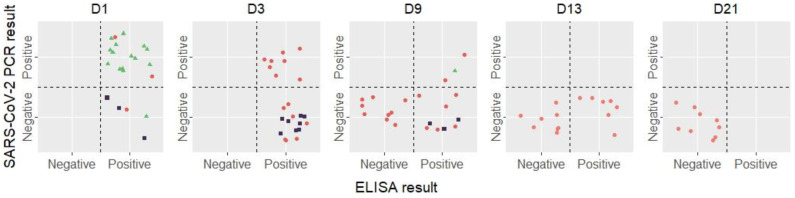
Evolution of RT-PCR and ELISA results of all samples during the study. The threshold for a positive ELISA is 3 pg/mL as recommended by the manufacturer. Red dots represent plasma samples, purple squares represent urine samples and green triangles represent respiratory samples.

**Figure 2 viruses-15-01041-f002:**
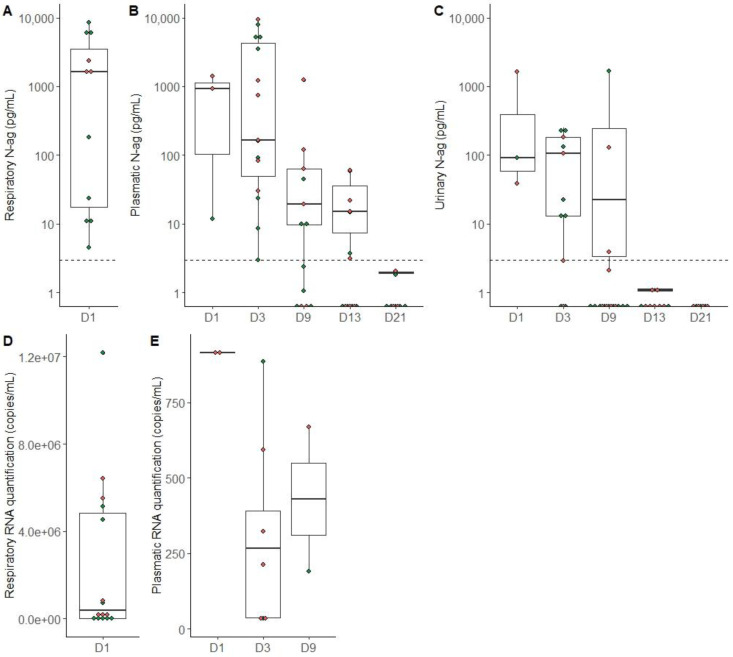
SARS-CoV-2 antigen and RNA quantification in different sample types according to the time since inclusion. Green dots represent patients who survived and red dots represent patients who died during hospitalization, and dotted lines represent the positivity threshold of the ELISA assay. (**A**) Antigen quantification in respiratory samples on day 1. (**B**) Antigen quantification in plasma samples from day 1 to day 21. (**C**) Antigenic quantification in urine from day 1 to day 21. (**D**) RNA quantification in respiratory samples on day 1. (**E**) RNA quantification in plasma samples from day 1 to day 9.

**Figure 3 viruses-15-01041-f003:**
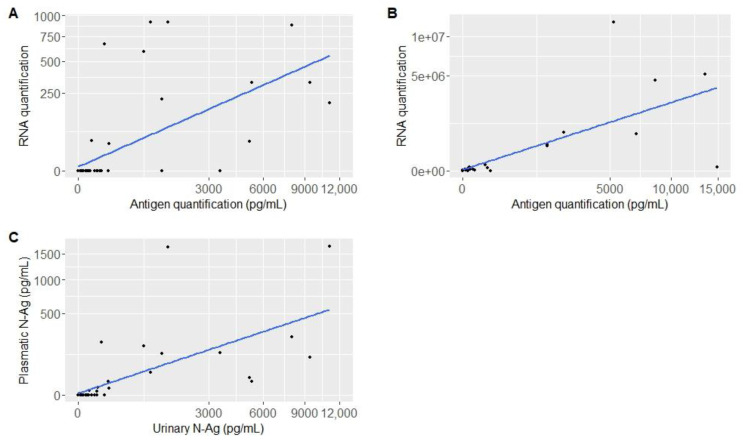
Representation of antigen quantification: according to RNA quantification for each patient in different compartments (**A**) in plasma, (**B**) in respiratory samples, and between compartments (**C**) in urine and in plasma.

**Table 1 viruses-15-01041-t001:** RT-PCR and ELISA results with quantification for all sample types, during time.

Plasma samples	N	D1, N = 3 ^1^	D3, N = 15 ^1^	D9, N = 18 ^1^	D13, N = 15 ^1^	D21, N = 9 ^1^
Positive RT-PCR	60	2 (67%)	8 (53%)	2 (11%)	0 (0%)	0 (0%)
RNA (copies/mL)	12	916 [916; 917]	267 [0; 886]	95 [0; 190]	NA	NA
Positive ELISA	59	3 (100%)	15 (100%)	8 (47%)	7 (47%)	0 (0%)
Antigen quantification (pg/mL)	59	919 [12; 1412]	165 [3; 9392]	2 [0; 11,108]	2 [0; 61]	0 [0; 2]
Urine samples	N	D1, N = 3 ^1^	D3, N = 12 ^1^	D9, N = 18 ^1^	D13, N = 14 ^1^	D21, N = 7 ^1^
Positive RT-PCR	14	0 (0%)	0 (0%)	0 (0%)	0 (%)	0 (%)
RNA (copies/mL)	0	NA	NA	NA	NA	NA
Positive ELISA	54	3 (100%)	8 (67%)	3 (17%)	0 (0%)	0 (0%)
Antigen quantification (pg/mL)	54	92 [38; 1651]	18 [0; 254]	0 [0; 1657]	0 [0; 1]	0 [0; 0]
Respiratory samples	N	D1, N = 16 ^1^		D9, N = 1 ^1^		
Positive RT-PCR	17	15 (94%)		1 (100%)		
RNA (copies/mL)	16	382,836 [210; 12,193,450]		335		
Positive ELISA	17	15 (94%)		1		
Antigen quantification (pg/mL)	17	2005 [0; 17,500]		3		

^1^ n (%); Median [Minimum; Maximum].

## Data Availability

The data presented in this study are available on request from the corresponding author.

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
