# Peer review of "SARS-CoV-2 N-Antigen Quantification in Respiratory Tract, Plasma and Urine: Kinetics and Association with RT-qPCR Results"

_viruses, 2023, doi:10.3390/v15051041_

Round 1

Reviewer 1 Report

This manuscript shows a quantitative study of SARS-CoV-2 N-antigen in several biological compartments and their correlations with viral RNA detection. It is an adequate manuscript based on thoroughly examined immunochromatography assay and RT-PCR methods. There are some ambiguities that I would like to discuss.

1. line 29: the authors said that “We detected N-antigen in urine and plasma samples until the second and third week after diagnosis”, but according to Table 2, the N-antigen was not detectable on day 21 in plasma samples and day 14 for urine samples. So this sentence was not rigorous and may gave wrong information.

2. line 108 and Table 2: The threshold for a positive result was 3 pg/ml in the N-antigen detection assay. Interestingly, the results of ELISA assay for respiratory samples on day 9 is exactly 3 and make the results “positive”. This coincidence should be brought in the manuscript.

3. line 226-229: The author said that the prolonged antigen shedding in both plasma and urine could be used in late diagnostic strategy. However, the positive rate is barely acceptable according to Table 2. Besides, the comparation should be carried out between ELISA on plasma (or urine) and RT-PCR on respiratory samples such as nasal swab or throat swab. So the potential of the antigen detection in urine is not that promising.

Author Response

  1. line 29: the authors said that “We detected N-antigen in urine and plasma samples until the second and third week after diagnosis”, but according to Table 2, the N-antigen was not detectable on day 21 in plasma samples and day 14 for urine samples. So this sentence was not rigorous and may gave wrong information. 

Response : We replaced it with the following sentence : "We detected N-antigen in urine and plasma samples until the day 9 and day 13 post-inclusion, respectively". It was an error in editing.

2. line 108 and Table 2: The threshold for a positive result was 3 pg/ml in the N-antigen detection assay. Interestingly, the results of ELISA assay for respiratory samples on day 9 is exactly 3 and make the results “positive”. This coincidence should be brought in the manuscript.

Response : The exact result was 3.07 pg/mL, we decided to round up to one significant figure. A sentence has been added to explain the coincidence.

3. line 226-229: The author said that the prolonged antigen shedding in both plasma and urine could be used in late diagnostic strategy. However, the positive rate is barely acceptable according to Table 2. Besides, the comparation should be carried out between ELISA on plasma (or urine) and RT-PCR on respiratory samples such as nasal swab or throat swab. So the potential of the antigen detection in urine is not that promising.

Response : We agree that it would have been interesting to compare antigen testing in respiratory samples to urine and plasma samples. Unfortunatly  respiratory specimens for these patients were not regularly collected (physicians focused on day 1 of inclusion for these samples). Nevertheless, as urine is an easy and painless sampel to obtain, we thought it would be interesting to highlight the idea of urinary testing in certain situations.  We removed the term "late" of the sentence, agreeing that it was the wrong term.

Reviewer 2 Report

In this manuscript the authors compare  SARS-CoV-2-PCR with SARS-CoV-2 antigen detection in different materials.

It is difficult to compare the value of the two methods for the course of infection as respiratory tract samples (if swabs or bronchial material is not declared) are only shown for the day of admission and not over time. Probably, testing of upper and lower respiratory tract samples over time by PCR is at least as  good as or even better than plasma/urine antigen testing. In addition, the number of patients and samples is far low. I am not convinced that an antigen ELISA has any benefit for the prognosis of the patients´ outcome.

Minor amendments:

I am missing the declaration that this is a retrospective study. There is no information about the storage of the samples (e.g. frozen).

In the introduction the authors speculate that antigen assays can help with triage. This idea is not discussed.

Most of the information in Table 1 is not described or discussed at all. Therefore, the information can be omitted.

Author Response

It is difficult to compare the value of the two methods for the course of infection as respiratory tract samples (if swabs or bronchial material is not declared) are only shown for the day of admission and not over time. Probably, testing of upper and lower respiratory tract samples over time by PCR is at least as  good as or even better than plasma/urine antigen testing. In addition, the number of patients and samples is far low. I am not convinced that an antigen ELISA has any benefit for the prognosis of the patients´ outcome.

Response : 

We totally agree with your comments, of course respiratory tract samples tested with PCR is better than antigen testing, and therefore to urine and plasma antigen testing. But still we think that urine is a really easy and painless sample to obtain, so we wanted to highlight antigen testing in this compartment. 

We do not have a high number of patients, due to lack of urine samples available in hospitalized patients. Indeed due to negative RNA detection in urine, physicians left the urinary detection aside and focused on other compartments. Thus, having a large group of patients was impossible for us.

Nothing is certain about prognosis elements discussed in the paper, they are just hints, but nevertheless it seemed important to us to publish data about urinary antigen detection and kinetics of excretion. Even if there is no direct clinical benefit today, this mayhelp in understanding the pathophysiology of this infection.

Minor amendments:

I am missing the declaration that this is a retrospective study. There is no information about the storage of the samples (e.g. frozen).

Response : Sentences have been added to specify that this is a retrospective study, and to specify the kind of storage.

In the introduction the authors speculate that antigen assays can help with triage. This idea is not discussed.

Response : this point is now discussed.

Most of the information in Table 1 is not described or discussed at all. Therefore, the information can be omitted.

Response : agreed, we added a sentence discussing certain parameters, and the table 1 will be a supplementary material.